# Age Differences in Attitudes towards Older People in Poland

**DOI:** 10.3390/ijerph18136869

**Published:** 2021-06-26

**Authors:** Elżbieta Krajewska-Kułak, Mateusz Cybulski, Paulina Aniśko, Magda Popławska

**Affiliations:** 1Department of Integrated Medical Care, Medical University of Białystok, 15-096 Białystok, Poland; mateusz.cybulski@umb.edu.pl (M.C.); paulinaanisko1@wp.pl (P.A.); 2Doctoral School, Medical University of Białystok, 15-089 Białystok, Poland; 3Students’ Scientific Society, Department of Integrated Medical Care, Medical University of Białystok, 15-096 Białystok, Poland; 20000227@wp.pl

**Keywords:** attitudes towards the older people, seniors, youth

## Abstract

Introduction: The attitudes towards older people is essential. Society’s perception of older adults is often unpleasant. Inappropriate treatment of older people in society causes a decrease in self-esteem, a sense of control over one’s own life, a decrease in the quality of life, an increase in anxiety, depression, anger, and sadness. The aim of this study was to compare the attitudes of young people and seniors towards older people in relation to their satisfaction with life and self-esteem. Materials and Methods: The study was conducted online in groups of young adults under 35 years of age (Group I) and over 65 years of age (group II), with 177 people each. Method: The study used the following questionnaires: Satisfaction with Life Scale (SWLS), Self-Assessment of Own Person (SOP), and Kogan’s Attitude Towards Old People Scale (KATOPS). Results: The satisfaction of life in both groups was average. The mean value of self-esteem was significantly (*p* < 0.001) lower in the group of seniors. Both groups of respondents had negative attitudes towards seniors in the KATOPS. There was a statistically significant correlation between life satisfaction and self-esteem in seniors. Gender, age, education, or place of residence of respondents had no impact on negative attitudes in both groups. Conclusions: Both groups showed negative attitudes towards seniors. No significant correlation was found between the groups and gender of the respondents, and the influence of the respondent’s age, education, marital status, or place of residence. There was a correlation between the respondents’ attitudes in the subscale of negative attitudes and their satisfaction with life. No correlation between self-esteem and age and marital status were found with no dependence on gender, education, or place of residence.

## 1. Introduction

In 2019, every fourth (25.3%) person in Poland was a senior—that is, 60 years old or older [1]. Over the years, in Poland and around the world, the image and lifestyles of seniors have undergone and continue to undergo significant changes. As a result, numerous stereotypes about seniors have developed in society, often based on myths and assumptions. Seniors are often thought of as unproductive, alienated, and ineffectual. Indeed, older people are often portrayed in the media and elsewhere as frail or weak, as suffering mental deterioration, and as poor and dependent. Parsons [2] has posited that society views the aged as an “increasing burden because they are unproductive, increasingly frail, and vulnerable with their decreasing ability to perform daily living activities and frequently poor mobility.” Since the last two decades of the twentieth century, however, the image of old age has especially changed [2,3]. Due to advances in medical care, more informed hygiene, better nutrition, more significant physical activity, better material conditions, and a higher level of education than the generations of their parents, an increasing number of people in late adulthood now enjoy relatively good health and live longer [4]. Although today’s seniors are often professionally, socially, and physically active, well-educated, creative people, they are rarely presented in that light. Today, many older people are active even after retirement. The term active ageing is often used. According to WHO, “active ageing is the process of optimizing opportunities for health, participation, and security in order to enhance the quality of life as people age” [4]. Active ageing applies to both individuals and population groups. It allows people to realize their potential for physical, social, and mental well-being throughout the life course and to participate in society. The phenomenon of stereotyping and the social marginalization of the older people and old age often persist, which can negatively affect the development of their self-esteem. Ageism is a type of discrimination that involves prejudice against people due to their age [5]. Similar to racism and sexism, ageism involves holding negative stereotypes about people of different ages. Gerontologist Robert N. Butler first used the term ageism to describe the discrimination of older adults. The term is often applied to any age-based discrimination, whether it involves prejudice against children, teenagers, adults, or senior citizens. Ageism is a highly prevalent and widespread phenomenon across many cultures. The literature emphasizes that young people do not attach importance to seniors’ experiences or knowledge. In their culture, the image of old age is inconsistent with reality and stereotypes [6,7,8,9,10]. Attitude theorists suggest that such attitudes are developed early in life and that, once formed, have enduring qualities that influence people’s thoughts and behavior throughout life [11]. Thus, in situations when negative stereotypes prevail, an attitude of discrimination against older adults arises—that is, either overt or subtle negative and/or unfair perceptions and treatment of older people, which can be especially harmful in efforts to protect their health and self-esteem [11,12].

Understanding attitudes towards older people is therefore essential. After all, self-esteem, defined as an individual’s subjective evaluation of their worth, is a vital aspect of the adaptive processes at all stages of life, particularly amongst older adults, being associated with the quality of the adaptation, well-being, satisfaction with life, and health [13]. According to terror management theory [14], self-esteem even has the effect of relieving negative emotions (e.g., the anxiety of mortality). Nevertheless, self-esteem has been rarely studied in older adults. In the limited literature on the topic, Borrico [15] has however demonstrated that self-esteem plays an important role in the attitudes of older adults, thereby confirming self-esteem’s significance in not only seniors’ social adaptation, but also their mental health. Stereotypical views on life and human development can promote the inappropriate treatment of seniors in society that may lower their self-esteem, sense of control over their lives, and their sense of quality of life [14,16]. Slevin [17] have found that negative perceptions of ageing and old age often exerts adverse impacts, because the emotional reaction generated by those perceptions in late adulthood deepens states of anxiety and fear as well as triggers depression, anger, and sadness, which can precipitate seniors’ withdrawal from life, loneliness, and sense of defeat and, consequently, shorten their lives. In recent studies on attitudes towards older adults [16,17,18], data have revealed that people hold more negative attitudes towards older adults than younger ones. Negative attitudes towards older people have indeed been reported in countries such as Ireland [17] and Sweden [18] with a high percentage of older people in the overall social structure. Researchers in Australia [19], Israel [20], and the United Kingdom [21] have made similar observations. By contrast, Alsenana’s [22] research revealed positive attitudes towards the older adults amongst respondents in China, Japan, India, and the Philippines—that is, in Eastern cultures, in which seniors are the center of attention in the family and both admired and respected.

Life satisfaction is defined as “a global subjective judgment concerning one’s life overall” [23], referring to a cognitive sense of satisfaction with life, life satisfaction involves a favorable attitude towards one’s life rather than an assessment of current feelings. There is strong evidence on the impact of other factors for life satisfaction such as marital status, education, employment, income, and being young [24,25].

Despite that work, little attention has been devoted to studying older people’s attitude perception in relation to their satisfaction with life and self-esteem. Therefore, our study aimed to evaluate whether intergenerational differences exist in the attitudes of young people and seniors towards older people and, if so, whether those attitudes are influenced by satisfaction with life and self-esteem. To that aim, the following research hypotheses were formulated:A relationship exists between self-esteem and attitudes towards older people;A correlation exists between satisfaction with life and attitudes towards older people;Attitudes towards older people differ between people of different ages; andThe gender, age, level of education, and place of residence of individuals impact their attitudes towards older people.

## 2. Materials and Methods

### 2.1. Procedure

A cross-sectional study was conducted from 1 June to 30 November 2019 with an online survey created using survey-creating software (survio.com). The respondents’ responses were recorded on the software’s platform Survio.com (SurveySparrow, Palo Alto, CA, USA) is a survey software for customer satisfaction, employee feedback, market research, and other online questionnaires. This software includes automatic response collection (start and end dates), statistics and graphs, downloaded as raw data, and prepared for statistical analysis. The average time to complete the questionnaire was 10 min. Inclusion criteria were: (1) age under 35 years of age and people over 65 years, (2) completing an online questionnaire, and (3) written consent to participate in the study. The exclusion criteria were: (1) age in the range of 36–64 years, (2) lack of possibility to fill out an online questionnaire, and (3) lack of written consent to participate. The survey was anonymous. Each participant was informed that they could withdraw from the study at any time. Once all questionnaires were returned, the completion rate was 47.3%, whereas 52.7% of the questionnaires were incomplete.

### 2.2. Measures

The Satisfaction with Life Scale (SWLS) measures the individual’s level of satisfaction with life at that moment in time [23]. It includes five items (i.e., “My life is close to ideal”, “The conditions of my life are excellent”, “I am satisfied with my life”, “I have gotten the important things I want in life”, and “If I could live my life over, I would change almost nothing”) to be rated on a 7-point Likert scale, ranging from 1 (strongly disagree) to 7 (strongly agree).

Scores were totaled within the range of 5 to 35 points; the higher the score, the greater the sense of satisfaction with life (31–35 points = extremely satisfied, 26–30 points = satisfied, 21–25 points = slightly satisfied, 20 points = neither satisfied nor dissatisfied, 15–19 points = slightly dissatisfied, 10–14 points = dissatisfied, 5–9 points = extremely dissatisfied). To interpret the results, the properties characterizing sten scores were also applied; sten scores ranging from 1 to 4 were treated as low, from 5 to 6 as average, and from 7 to 10 as high. The reliability index (Cronbach’s alpha) of the SWLS, established in a study of 371 adults, is 0.81. The SWLS was also demonstrated to be valid amongst Polish participants [23].

The Self-Assessment Scale of Own Person is a Polish scale in which respondents assess themselves [26]. The instrument has 32 items, which respondents rate on a 5-point scale for a corresponding number of points (4 points = very often, 3 points = often, 2 points = sometimes, 1 point = rarely, 0 points = never). For each respondent, the points earned were totaled for overall score of the degree of self-esteem in a range from 0 to 128 points; the higher the score, the lower the self-esteem (0–25 points = overestimated self-esteem, 26–45 points = adequate self-esteem, 46–128 points = understated self-esteem).

The Polish version of the Kogan Attitudes toward Older People Scale (KAOPS) [23,27] is a 34-item self-report scale designed to measure older adults’ attitudes. The 17 negatively worded items form the Older Person Negative subscale, whereas the other 17, all positively worded, form the Older Person Positive subscale. The items are rated on a 6-point summed Likert response attitude measure ranging from 1 (strongly disagree) to 6 (strongly agree), with 4 points given in the event that the respondent failed to rate an item [16]. Overall scores on the KAOPS range from 34 to 204 points, with the borderline score between positive and negative attitudes of 119 points, with higher scores indicating a more positive attitude. The questionnaire consists of two subscales—the positive attitude subscale and the negative attitude subscale—each of which awards from 17 to 85 points. The higher the score on each subscale, the greater the intensity of the positive or negative attitude. To facilitate the interpretation of the KAOPS, the following categories were used: 34–118 points = high negative attitude, 119–146 = low negative attitude, 147–174 = low positive attitude, 175–204 = high positive attitude. Cronbach’s α coefficient for the questionnaire is 0.81.

### 2.3. Participants

The study was performed with a sample of 177 randomly selected adults less than 35 years old (Group I) and 177 randomly selected adults more than 65 years old (Group II). This study recruited individuals across Poland via social media to complete an online survey. Study advertisements and the study website were shared via Facebook.

In the young people group, with 148 women (83.6%) and 29 men (16.4), the mean age was 24.4 ± 4.5 years, the youngest respondent was 16 years old, and the oldest was 35 years old. In the senior Group, with 141 women (79.7%) and 36 men (20.3%), the mean age was 68.9 ± 4.9 years, the youngest respondent was 66 years old, and the oldest was 85 years old. Details about the respondents in each Group appear in Table 1.

### 2.4. Statistical Analysis

Once the data were collected in a Microsoft Excel 2013 (Microsoft Corporation, Albuquerque, NM, USA) spreadsheet, statistical analysis was performed in Statistica 13.0 (StatSoft Company, Hamburg, Germany) PL. Data are expressed as mean and standard deviation. Differences between young adult and seniors groups were analyzed by the Mann–Whitney test. For correlations, Spearman *r* and variance analysis is reported.

The results were considered statistically significant at *p* < 0.05.

## 3. Results

The average degree of satisfaction with life in Group I was 23.2 ± 6.5 points, and in Group II was 21.8 ± 5.1 points (Table 2). Details are shown in Appendix A.

Those findings indicate that young respondents and seniors were relatively satisfied with their lives, although respondents in Group I were more satisfied (*p* = 0.03). After converting the obtained results into sten according to the manufacturer’s recommendations, the mean value in Group I was 6.6 ± 2.3 sten, and in Group II was 5.96 ± 1.8, which confirmed the average satisfaction with life in both groups, with a higher tendency in the group of young people (*p* = 0.004).

The average value of self-esteem in Group I was 66.6 ± 22.2, and in Group II was 85.01 ± 28.3. Those results indicate that in both groups, self-esteem was low, although lower in the Group of seniors (*p* < 0.001). Details about the results of the Self-Assessment Scale of Own Person appear in Table 3 and Appendix A.

On the KAOPS, respondents in Group I scored 118.7 ± 16.5 points on average, whereas ones in Group II averaged 115.5 ± 11.4 points. Details about the results of the KAOPS appear in Appendix A. Both groups showed negative attitudes towards seniors. No significant correlation between the groups and the gender of respondents in preferred attitude towards seniors was found. The respondents’ age (*p* = 0.004), level of education (*p* < 0.01), marital status (*p* = 0.006), and place of residence *(p* = 0.02) influenced attitudes toward older people.

Concerning the attitudes divided according to the subscales of negative and positive attitudes, Group I represented the negative attitude subscale, with respondents scoring 61.6 ± 8.8 points on average (min. = 36, max. = 79), and Group II, scoring 56.97 ± 8.2 points on average (min. = 43, max. = 86), represented the positive attitudes subscale (*p* < 0.001).

No statistically significant correlation emerged between the preferred values on the subscale of negative attitudes between respondents from Groups I and II, nor was there any influence due to their gender, age, level of education, or place of residence. Only marital status influenced the results for the negative attitude subscale (*F* = 4.702, *p* = 0.01). There was also no correlation between the results for the subscale of negative attitudes in Groups I and II, their self-esteem, and the influence of their gender and place of residence, whereas relationships between the groups and age (*F* = 3.703, *p* = 0.03) and marital status (*F* = 3.434, *p* = 0.03) did emerge. There was also a correlation between the attitudes of the respondents in both groups on the subscale and their satisfaction with life (*R* = 0.189, *p* = 0.01), depending on the age of respondents (*F* = 4.729, *p* = 0.01) and their marital status (*F* = 4.332, *p* = 0.01), but not their gender, level of education, or place of residence. No correlation was found between the respondents’ attitudes on the subscale and their self-esteem, and no relationship between the respondents’ attitudes on the subscale and gender, level of education, or place of residence was noted. However, those results did depend upon age (*F* = 7.653, *p* < 0.001) and marital status (*F* = 3.869, *p* = 0.02).

On the positive attitude subscale, respondents scored average of 56.7 ± 9.1 points in Group I and 58.5 ± 7.7 points in Group II (*p* = 0.04). The data support the hypothesis that negative attitudes toward older adults dominate amongst younger adults. Positive attitudes correlated significantly with satisfaction with life (*R* = 0.189, *p* = 0.01), but was not influenced by gender, level of education, or place of residence, despite being influenced by age (*F* = 4.729, *p* = 0.01) and marital status (*F* = 4.332, *p* = 0.01).

Concerning the attitudes divided according to the subscales of negative and positive attitudes, Group I represented the negative attitude subscale, with respondents scoring 61.6 ± 8.8 points on average (min. = 36, max. = 79), and Group II, scoring 56.97 ± 8.2 points on average (min. = 43, max. = 86), represented the positive attitudes subscale (*p* < 0.001). Details are included in Table 4.

## 4. Discussion

The aim of our study was to compare the attitudes of young people and seniors towards older adults in relation to their satisfaction with life and self-esteem. We found that adults less than 35 years of age and over 65 years of age were relatively satisfied with their lives, although younger people were more satisfied. Beyond that, self-esteem was lower in the group of seniors. Both groups showed negative attitudes towards seniors, and those attitudes depended on their satisfaction with life. In fact, the analysis of results obtained from the KAOPS showed that both groups held rather negative attitudes towards older people, although young people had more negative attitudes than seniors.

Several factors may influence life satisfaction. There is strong evidence on the influence factors such as age, marital status, education, employment, and income [25]. Our findings are in agreement with previous studies [25,28,29,30]. Luchesi et al. [28] have demonstrated that more negative attitudes to the older adults were associated with life satisfaction. Also, living in an urban setting, taking more medications per day, and dependent on basic daily living activities were correlated with negative attitudes to the older adults. In an Australian study [29], people over 60 years have reported relatively high satisfaction with life, scoring on average 26.63 out of 35 on SWLS. Furthermore, age did not correlate with satisfaction with life, and there was no gender difference. More positive attitudes to ageing were associated with higher satisfaction levels with life, better self-reporting of physical and mental health. Also, participants in the Berlin Aging Study were described as largely very satisfied with life, even when living with high levels of functional limitation [30]. In the present study, self-esteem was lower in older people. This result was consistent with the results from previous studies [31,32]. Self-esteem is linked to well-being, life satisfaction, health, and not related to chronological age [31]. Hunter et al. [32] demonstrated that the low self-esteem group had poorer self-reported health, more pain, higher disability, and had significantly higher scores on depression and anxiety.

In the ageing process, as physical functioning consistently declines, the situation surrounding an older person changes remarkably due to economic loss, one’s own life expectancy, and separation from one’s spouse and relatives [13,14,15,23]. Those changes are considered to influence the quality of life and satisfaction with life. Ageing also increases negative attitudes towards older people. In particular, a negative attitude towards one’s own ageing tends to be associated with poorer subjective health, lower satisfaction with life, and other indicators of diminished functioning [23].

Several studies conducted on attitudes towards older people in Poland have shown different results. The research of Trempa and Zając-Lamparska [5] revealed that students had neutral attitudes towards older people and old age. The results obtained by Miłkowska [8], by comparison, suggest that students held positive and negative attitudes towards old age, but positive attitudes towards specific older people, namely their grandparents. Amongst other results, respondents in Cybulska et al. [33] study had positive attitudes towards ageing. Similar results have been reported by other authors [19,20,21,22,23,34,35]. Added to that, in studies conducted in Poland [8,21,36,37,38,39,40], young people’s attitudes towards the older people have been found to be influenced by stereotypes, with older people often viewed as unattractive, sick, intellectually limited, and unhappy.

In our study, young respondents and seniors were relatively satisfied with their lives, while younger respondents were more satisfied. Self-esteem was lower amongst seniors, and positive attitudes towards older people were correlated with satisfaction with life. In the study by Strugała [41], differences in self-esteem and perceptions of old age between the study groups (i.e., >60 years old and <60 years old) were slight.

Important observations have also been presented by Nierzewska and Gurba [42], who found more positive attitudes and less intense negative attitudes amongst people with broader knowledge about old age. Other studies have also indicated such a relationship [43,44,45].

No correlations between negative attitudes towards older people and self-esteem and satisfaction with life as well as positive attitudes and self-esteem were found. A statistically significant relationship between the groups existed only between the posed positive attitudes and perceived satisfaction with life. We suggest that a relatively small group of respondents, an average satisfaction with life, and an average self-esteem may explain the lack of significant correlations.

The level of knowledge about old age allows predicting the intensity of positive and negative attitudes towards older people and is the strongest predictor of those attitudes. The intensity of positive attitudes is also predicted by the subjective assessment of the quality of contact with older people [36,46,47,48,49,50]. In view of those results, Zawada [51] has stressed that gerontological education should be included in education programs from kindergarten to university. Such knowledge would contribute to the dismantling of stereotypes about older people and old age and the development of positive attitudes towards seniors. According to other research [8,47,49], to eliminate current stereotypes towards old age, integrated social actions—for example, the early education of society and media’s popularization of knowledge about old age and older people—should be undertaken as soon as possible. Knowledge about ageing, old age, and the quality of life of seniors is necessary to properly recognize seniors’ expectations and needs, thereby enabling the creation of programs to stimulate the development of seniors and improve social attitudes, making the last stage of human life more satisfactory.

In growing older, people can become increasingly vulnerable to the effects of ageism [5]. Active ageing means helping people stay in charge of their own lives for as long as possible as they age and, where possible, to contribute to the economy and society. For example, European programs such as Employment and Social Innovation and the European Social Fund have funded projects to support employment, social policy, and labor mobility of older people [4].

On the other hand, ageism can theoretically affect all individuals who enter older age [52]. Discrimination against older people can be a lack of employment opportunities and career progression.

Moreover, other forms of discrimination against older people may be biased decision-making in healthcare, mental health, and general care [53].

Our findings have some potential limitations. First, the study group was too small to generalize the results to the entire population of young people and seniors in Poland. Second, there was an overrepresentation of women (83.6%) in the studied groups; thus, the results should be verified in a sample with equal numbers of men and women. Third, our study’s variables were assessed via self-reported data online, although the questionnaire survey method can measure only individuals’ subjective attitudes and may be affected by social desirability bias. The Implicit Association Test, by contrast, which measures individuals’ implicit attitudes, has been shown to avoid the information distortion caused by self-presentation so that respondents report their true attitudes. Fourth, we used only three scales, namely to assess satisfaction with life, the self-esteem, and attitudes towards older people.

Despite those limitations, our study’s results may constitute a starting point for further research on attitudes towards old age and its perception among young people and seniors. Ideally, those results should be verified in a longitudinal study. In the future, we plan to expand the number of respondents sampled and have a similar number of women and men. We also plan to use traditional questionnaires and include respondents 36 to 64 years old.

## 5. Conclusions

Both young adults and older adults showed negative attitudes towards seniors.No significant correlation between the groups in terms of gender, age level of education, marital status, or place of residence was found.There was a correlation between the respondents’ attitudes on the subscale of negative attitudes and their satisfaction with life.No correlations between the respondents’ attitudes toward older people on the subscale and their self-esteem and age, and marital status were found that depended on gender, level of education, or residence.

## Figures and Tables

**Table 1 ijerph-18-06869-t001:** Demographic data of the studied groups.

Variable	Young Up to 35 Years Old	Seniors Over 65 Years of Age
Age in years		
Mean (SD), Median (q1–q3)	24.4 ± 4.5; 24 (20–27)	68.9 ± 4.9; 69 (65–72)
Minimum	16	66
Maximum	35	85
Gender		
Woman	83.6%	79.7%
Man	16.4%	20.3%
Place of residence		
Village	17.5%	6.8%
City with up to 100,000 inhabitants	22.6%	7.3%
City with up to 200,000 inhabitants	5.1%	7.9%
City with up to 300,000 inhabitants	38.4%	35.7%
City with more than 300,000 inhabitants	16.4%	24.2%
Level of education		
Secondary	3.4%	22.6%
Bachelor	53.1%	74.4%
University	42.9%	
Vocational		3%
Marital status		
Married	41.8%	55.4%
Single	40.1%	10.2%
In an informal relationship	18.1%	1.7%
Divorced		9.6%
Widowed		23.2%

**Table 2 ijerph-18-06869-t002:** Results of the Satisfaction with Life Scale (SWLS) of young people and seniors.

**The Average Degree (SD), Median (q1–q3) of Satisfaction with Life in Points**	**Group I**	**Group II**	***p*-Value**
	**23.2 ± 6.5; 24 (20–28)**	**21.8 ± 5.1; 22 (19–25)**	**0.03**
Points minimum-maximum	5–35	10–34	
The obtained results of satisfaction with life into sten mean (SD), Median (q1–q3)	6.6 ± 2.3; 7 (5–8)	5.96 ± 1.8; 5 (5–7)	0.004
Sten minimum-maximum	1–10	2–10	

SD-standard deviation; q-quartile.

**Table 3 ijerph-18-06869-t003:** Results according to the Self-Assessment Scale of Own Person.

**Average of the Points Mean (SD); Median (q1–q3)**	**Group I**	**Group II**	***p*-Value**
	**66.6 ± 22.2; 57 (54–61)**	**85.01 ± 28.3; 65 (51–80)**	**<0.001**
Points minimum–maximum	26–121	52–117	

SD-standard deviation; q-quartile.

**Table 4 ijerph-18-06869-t004:** The attitudes of respondents towards the older adults according to the Kogan scale.

Attitudes	Average Scores by Kogan		*p*-Value
	Young People No = 177 Mean (SD);Median (q1–q3)	Seniors No = 177 Mean (SD); Median (q1–q3)	
**Negative attitudes**	61.6 ± 8.8; 61(56–67)	56.97 ± 8.2; 57 (54–61)	<0.001
**Positive attitudes**	56.72 ± 9.13; 57 (51–62)	58.51 ± 7.72; 59 (54–63)	<0.05

SD-standard deviation; q-quartile.

## Data Availability

Data are available upon reasonable request.

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
