# Peer review of "Age Differences in Attitudes towards Older People in Poland"

_ijerph, 2021, doi:10.3390/ijerph18136869_

Round 1

Reviewer 1 Report

The reviewed article is interesting for the reader. The topic is relevant to today's society. Intergenerational relationships are very well presented in the article. It is a well-written manuscript that provides new information. The authors successfully achieved the aim of the study. They used a representative group and an interesting research methodology. The results and discussion opened a new approach to the interpretation of data on this subject. The conclusions fully reflect the obtained results and their interpretation.

Author Response

Thank you for positive review our article. We will check and correct English language.

Elżbieta Krajewska-Kulak

Reviewer 2 Report

The present manuscript is about age differences in attitudes among older adults. It seems to be an interesting paper. However, I want to give some suggestions to the authors:

  1. First, please use "older adult", "older people" or similar instead of "elderly", because of the negative connotation.
  2. The abstract is too long. I suggest the authors reduce the results on the abstract and explain more about method. 
  3. English editing is required along the manuscript. You can use the English editing of MDPI or a particular, but it is necessary. 
  4. Lines 40-43. Please provide the page where you get the information "increasing burden on society because they are unproductive, increasingly frail, and vulnerable with their decreasing ability to perform daily living activities and frequently poor mobility." 
  5. In the introduction I miss the authors talk about ageism and active aging. They are not mentioned we discuss even though related topics. 
  6. Lines 97-102. Format mistakes. Please review.
  7. I would like them to better clarify why they select the group of "young adults" who are under 35 years of age, why under 35 years of age, and on what basis you establish 35 years as the limit. Do you think that the opinion of a 16-year-old teenager differs greatly from that of a 35-year-old? I don't quite understand the selection of that age range or the purpose.
  8. Table 1: Are there big differences in populations of 100, 200, or 300 thousand people? I think it would be more interesting to make an evidence-based classification. On seniors when you explain the mean age you put "lat" at the end. Is it a mistake or does it mean something? In education, what do you mean by professional? One thing is the level of studies and another thing is the work activity, they are mixing concepts. What is "mingle"? "Informal relationship"? 
  9. Statistical analysis: the authors must explain in more detail the analysis. The variables included in the different type of analysis, provide tables with these data. The authors mentioned some details in the text, but you have to provide more data about it. I don't understand the way you explain the results. Where is the regression? You only put this: "The data support the hypothesis that negative attitudes dominated group I. Positive attitudes correlated significantly with life satisfaction (R = 0.189, p = 0.011)."
  10. Line 188. When the p-value is over 0.05 you can put "p>0.05". 
  11. Table 3 is huge and incomprehensible. The most important is the average of points, minimum and maximum. Please explain better the table and think about the structuration of the table and the key information. 
  12. Table 4. "No" must be "n" in lowercase because N refers to the total sample. Think about separate negative and positive attitudes in different tables. 
  13. Please consider the suggestions and clarify the results and give details about correlations and regressions with the size of the effect and statistical power. 
  14. The discussion section is correct. Please add ageism and active aging in a brief paragraph because as they are relevant aspects to mention and in the introduction section (as aforementioned). 

Author Response

  • First, please use "older adult", "older people" or similar instead of "elderly", because of the negative connotation.

We have changed elderly into older people.

  • The abstract is too long. I suggest the authors reduce the results on the abstract and explain more about method. 

We have reduced the results section in the abstract.

  • English editing is required along the manuscript. You can use the English editing of MDPI or a particular, but it is necessary. 

English has been corrected. See, the corrected language.

  • Lines 40-43. Please provide the page where you get the information "increasing burden on society because they are unproductive, increasingly frail, and vulnerable with their decreasing ability to perform daily living activities and frequently poor mobility." 
  • In the introduction I miss the authors talk about ageism and active aging. They are not mentioned we discuss even though related topics. 

We have provided about ageism and active aging.

  • Lines 97-102. Format mistakes. Please review.

We have corrected it.

  • I would like them to better clarify why they select the group of "young adults" who are under 35 years of age, why under 35 years of age, and on what basis you establish 35 years as the limit. Do you think that the opinion of a 16-year-old teenager differs greatly from that of a 35-year-old? I don't quite understand the selection of that age range or the purpose.

In the this study we have tried

In Polish scientific literature, the term young adult under the age of 35 is used.

  • Table 1: Are there big differences in populations of 100, 200, or 300 thousand people? I think it would be more interesting to make an evidence-based classification. On seniors when you explain the mean age you put "lat" at the end. Is it a mistake or does it mean something?

This is mistake. We have corrected it.

In education, what do you mean by professional? One thing is the level of studies and another thing is the work activity, they are mixing concepts. What is "mingle"? "Informal relationship"? 

We have changed the word professional into vocational.

  • Statistical analysis: the authors must explain in more detail the analysis. The variables included in the different type of analysis, provide tables with these data. The authors mentioned some details in the text, but you have to provide more data about it. I don't understand the way you explain the results. Where is the regression?

We have provided more details on statistical analysis. We have used  analysis of correlation  not regression.

You only put this: "The data support the hypothesis that negative attitudes dominated group I. Positive attitudes correlated significantly with life satisfaction (R = 0.189, p = 0.011)."

  • Line 188. When the p-value is over 0.05 you can put "p>0.05". 

We have corrected it.

  • Table 3 is huge and incomprehensible. The most important is the average of points, minimum and maximum. Please explain better the table and think about the structuration of the table and the key information. 

We agree that Table 3 is huge. But only average points, minimum and maximum do not explain the essence of this scale.  Maybe this table can be placed as additional material.

  • Table 4. "No" must be "n" in lowercase because N refers to the total sample. Think about separate negative and positive attitudes in different tables. 

We have corrected “No” to lowercase.

  • Please consider the suggestions and clarify the results and give details about correlations and regressions with the size of the effect and statistical power. 

We have used analysis of  correlation not regression.

  • The discussion section is correct. Please add ageism and active aging in a brief paragraph because as they are relevant aspects to mention and in the introduction section (as aforementioned). 

We have added on ageism and active aging in the Discussion section.

In growing older, people can become increasingly vulnerable to the effects of ageism [5].  Active ageing means helping people stay in charge of their own lives for as long as possible as they age and, where possible, to contribute to the economy and society. For example, European programmes such as Employment and Social Innovation and the  European Social Fund have funded projects to support employment, social policy, and labour mobility of older people [4].

On the other hand, ageism can theoretically affect all individuals who enter older age [40]. Discrimination against older people can be a lack of employment opportunities and career progression.

Moreover, other forms of discrimination against older people may be biased decision-making in healthcare and mental health and general care [41]. 

Reviewer 3 Report

Thank you for recommending me as a reviewer. This paper was to aim com-13 pare the attitudes of young people and seniors towards the elderly in relation to their satisfaction 14 with life and self-esteem. If the authors complete minor revisions, the quality of the study will be further improved. 1. Abstract: Authors need to make sure that the form "Abstract: Introduction:" is suitable. 2. The introduction section is well written. However, there are too many paragraphs. I suggest in the introduction section that the author combines several paragraphs into one paragraph. 3. line 105-107: "The cross-sectional study was conducted from June 1 to November 30, 2019. The 105 study was conducted based on an online survey created using dedicated software (sur-106 vio.com)." - The author needs to be more specific about "sur-106 vio.com". 4. line 114-115: "Concerning the total number of questionnaires returned, the complete completion rate was 47.3%. The remaining percentage (52.7%) is incomplete surveys." - In this study, the rate of non-response of the survey was too high. Authors should add to the limitations of the discussion section why the non-response rate is high. In addition, if necessary, authors may need to perform an additional 'analysis of missing value ' in the results section. The authors can decide if further analysis is necessary. 5. In this study, the author can change the method of marking the significance level. ex. p = 0.0266 -> p = 0.03 6. The discussion section also has too many paragraphs. Authors may combine some paragraphs of the discussion section into one paragraph. 5.

Author Response

Thank you for recommending me as a reviewer. This paper was to aim com-13 pare the attitudes of young people and seniors towards the elderly in relation to their satisfaction 14 with life and self-esteem. If the authors complete minor revisions, the quality of the study will be further improved.

  1. Abstract: Authors need to make sure that the form "Abstract: Introduction:" is suitable.

        Negative attitudes toward older people in society cause a decrease in the quality of life, an increase in anxiety, depression, anger, and sadness.

  1. The introduction section is well written. However, there are too many paragraphs. I suggest in the introduction section that the author combines several paragraphs into one paragraph.

The introduction section paragraphs were combined into one paragraph.

  1. line 105-107: "The cross-sectional study was conducted from June 1 to November 30, 2019. The 105 study was conducted based on an online survey created using dedicated software (sur-106 vio.com)." –

 The author needs to be more specific about "sur-106 vio.com". 4. line 114-115:

Survio.com is a survey software for customer satisfaction, employee feedback, market research, and other online questionnaires. This software includes automatic response collection (start and end dates), statistics and graphs,

"Concerning the total number of questionnaires returned, the complete completion rate was 47.3%. The remaining percentage (52.7%) is incomplete surveys." - In this study, the rate of non-response of the survey was too high. Authors should add to the limitations of the discussion section why the non-response rate is high. In addition, if necessary, authors may need to perform an additional 'analysis of missing value ' in the results section. The authors can decide if further analysis is necessary. 5. In this study, the author can change the method of marking the significance level. ex. p = 0.0266 -> p = 0.03 6.

We have changed with the suggestion.

The discussion section also has too many paragraphs. Authors may combine some paragraphs of the discussion section into one paragraph. 5.

The discussion section paragraphs were combined into one paragraph.

Reviewer 4 Report

I recommend the following revisions:

  1. To further explicit the relevance of the study of self-esteem and life satisfaction to attitudes towards older people (introduction)
  2. To explain how study participants were recruited and selected;
  3. In what concerns inclusion and exclusion criteria, clarify if it was ensured that the study participants had the cognitive capacity to answer the questionnaire;
  4. To mention if the scales "Satisfaction with Life Scale" and "Self-Assessment Scale of Own Person" are validated for Poland;
  5. To revise the format of table 2 e 3 to obtain a better data presentation and comprehension;
  6. To review the conclusions to promote a complete connection between the study's purpose and its findings (a relationship between self-esteem and attitudes towards older people) and drawing implications for research and practice.

Author Response

  • To further explicit the relevance of the study of self-esteem and life satisfaction to attitudes towards older people (introduction)

We have provide several sentences on self-esteem (Introduction section)

Understanding attitudes towards older people is therefore essential. After all, self-esteem, defined as an individual's subjective evaluation of their worth, is a vital aspect of the adaptive processes at all stages of life, particularly amongst older adults, one associated with the quality of the adaptation, well-being, satisfaction with life, and health [13]. According to terror management theory [14], self-esteem even has the effect of relieving negative emotions (e.g., the anxiety of mortality). Nevertheless, self-esteem has been rarely studied in older adults. In the limited literature on the topic, Borrico [15] has however demonstrated that self-esteem plays an important role in the attitudes of older adults, thereby confirming self-esteem's significance in not only seniors' social adaptation but also their mental health. Stereotypical views on life and human development can promote the inappropriate treatment of seniors in society that may lower their self-esteem, sense of control over their lives, and their sense of quality of life [14,16].

Life satisfaction is defined as “a global subjective judgment concerning one’s life overall”[24 ], referring to a cognitive sense of satisfaction with life, Life satisfaction involves a favorable attitude towards one's life rather than an assessment of current feelings. There is strong evidence on the impact of other factors for life satisfaction such as marital status, education, employment, income, and being young  [25].

  • To explain how study participants were recruited and selected;

This study recruited individuals across Poland via social media to complete an online

survey. Study advertisements and the study website were shared via

Facebook. (Materials and methods, Participants )

  • In what concerns inclusion and exclusion criteria, clarify if it was ensured that the study participants had the cognitive capacity to answer the questionnaire;

Inclusion criteria were : (1). age under 35 years of age and people over 65 years, (2). Completing an online questionnaire, (3) written consent to participate in the study. The exclusion criteria were:  (1) Age in the range of 36-64 years,( 2) lack of possibility to fill out an online questionnaire, and  and (3) lack of written consent to participate. . (Materials and methods, Procedure)

  • To mention if the scales "Satisfaction with Life Scale" and "Self-Assessment Scale of Own Person" are validated for Poland;

The scales Satisfaction with Life Scale and Self-Assessment Scale of Own Person were validated for Poland. The SWLS was also demonstrated to be valid amongst Polish participants [23]. (Materials and methods).

The Self-Assessment Scale of Own Person is a Polish scale in which respondents assess themselves [26]. (Materials and methods).

  • To revise the format of table 2 e 3 to obtain a better data presentation and comprehension;

These tables have a lot of data, so we were unable to present data in another way. Maybe some solution it  will be to include these tables in the supplemental files.

  • To review the conclusions to promote a complete connection between the study's purpose and its findings (a relationship between self-esteem and attitudes towards older people) and drawing implications for research and practice.

No correlations between the respondents' attitudes toward older people on the subscale and their self-esteem and age, and marital status were found that depended on gender, level of education, or residence. (Conclusion section)

 Please find the corrected manucsript.

Reviewer 5 Report

This article is interesting and should be rewritten and resubmitted for review. Specifically, the following is recommended:

  1. Authors should consult with an English native to correct language mistakes. There are quite a few (e.g. To summing up..). Some sentences are not finished (e.g. Third, given that our study variables were assessed via self-reported data online”) or written in colloquial English (e.g. Getting to know…). Please look at different articles to see how socio-demographic variables are properly named (e.g. the word “mingle” or “professional”). “The borderline point”, I guess the authors meant the cut-off point.
  2. Please avoid the term “elderly”. It is outdated and discriminatory. You can find this information on many gerontological journals’ websites.
  3. The title of the article must be changed. The authors present data on Polish population and the title should include this information.
  4. A paragraph should not consist of only one sentence. I suggest the authors rewrite this section and combine sentences to improve the quality of writing. Please also work on coherence (e.g. “Understanding attitudes towards older people is essential. Self-esteem is an ….).
  5. What is the point of creating such a big table with all the items listed if the items are not analyzed individually?
  6. The discussion section needs to be rewritten. The authors should be more specific about the implications of the study.

Author Response

  1. Authors should consult with an English native to correct language mistakes. There are quite a few (e.g. To summing up..). Some sentences are not finished (e.g. Third, given that our study variables were assessed via self-reported data online”) or written in colloquial English (e.g. Getting to know…). Please look at different articles to see how socio-demographic variables are properly named (e.g. the word “mingle” or “professional”). “The borderline point”, I guess the authors meant the cut-off point.

The article has been corrected by English native speaker (EditMyEnglish). We have included English correction

  1. Please avoid the term “elderly”. It is outdated and discriminatory. You can find this information on many gerontological journals’ websites.

We have changed term elderly into older people

  1. The title of the article must be changed. The authors present data on Polish population and the title should include this information.

We have changed the title. Age differences in attitudes towards older people in Poland

  1. A paragraph should not consist of only one sentence. I suggest the authors rewrite this section and combine sentences to improve the quality of writing. Please also work on coherence (e.g. “Understanding attitudes towards older people is essential. Self-esteem is an ….).

English has been corrected.

  1. What is the point of creating such a big table with all the items listed if the items are not analyzed individually?

We have created such big table to provide more details.

  1. The discussion section needs to be rewritten. The authors should be more specific about the implications of the study. We have added several sentences in the Discussiin section.

Several factors may influence life satisfaction. There is strong evidence on the influence factors such as age, marital status, education, employment, and income [25]. Our findings are in agreement  with previous studies [25,28,29,30]. Luchesi et al.  [28] have demonstrated that more negative attitudes to the elderly were associated with life satisfaction. Also, living in an urban setting, taking more medications per day, and dependent on basic daily living activities were correlated with negative attitudes to the elderly. In an Australian study [29], people over 60 years have reported relatively high satisfaction with life, scoring on average 26.63 out of 35 on SWLS. Furthermore, age did not correlate with satisfaction with life, and there was no gender difference. More positive attitudes to aging were associated with higher satisfaction levels with life, better self-report physical and mental health.  Also, participants in the Berlin Aging Study were described as largely very satisfied with life, even when living with high levels of functional limitation  [30]. In the present study, self-esteem was lower in older people. This result was consistent with the results from previous studies [31,32]. Self-esteem is linked to well-being, life satisfaction, health and not related to chronological age [31]. Hunter et al. [32] have demonstrated that the low self-esteem group had poorer self-reported health, more pain, higher disability, and had significantly higher scores on depression and anxiety.

Round 2

Reviewer 2 Report

Thank you for your revised version.  However, I have to give you more recommendations to consider the present paper to be published.

  1. Abstract: In the first line you mentioned "public", it's better to use the term "society".
  2. The introduction is adequate. However, the authors need to review English writing along the section and the manuscript.
  3. Material and methods. Procedure. Did you use an anonymous survey? Why did you include "written consent to participate in the study" if it is an online survey? How did you obtain the written consent? How did you contact and select the participants?
  4. Material and methods. Measures. "It includes five items (i.e.)...", why did you put i.e. if you mentioned the five items?
  5. In the following phrase: "To
    interpret the results, the properties characterising sten scores were also applied; sten". What did you refer to "sten"?
  6. Table 1: it's better to put a range (minimum, maximum) in the same row. For example: Age of young up to 35 years old (16, 35).
  7. Table 2. I don't understand the reason to put frequencies of all the points (1 to 7). A mean and standard desviation of each statement it's better. and more comprehensive as in the case by group I and II. What is it sten? I don't understand. 
  8. Table 3: The same as table 2. You have you provide the mean and standard desviation. The tables are so huge. 
  9. Table 4: why did you provide so much information? Can you divide into negative and positive attitudes in different tables, please? Or please, look for a simple way to understand the table.
  10. Please English writing must improve. 

Author Response

  1. Abstract: In the first line you mentioned "public", it's better to use the term "society".

We have changed it ( Abstract section the first line)

  1. The introduction is adequate. However, the authors need to review English writing along the section and the manuscript.
  2. The article and the last changes in the Introduction section have been corrected by  (EditMyEnglish).

         Please see the corrected Introduction the last changes and all manuscript

  1. Material and methods. Procedure. Did you use an anonymous survey? The survey was anonymous.  (Materials and methods, Procedure)
  2. Why did you include "written consent to participate in the study" if it is an online survey? This was a requirement of the Bioethics Committee of the Medical University of Bialystok

How did you obtain the written consent? We have used electronic consent in the survey: Please select your choice below. Clicking on the “Agree” button indicates that  You have read the study information. You voluntarily agree to participate.  ¨  Agree,  ¨  Disagree

  1.  How did you contact and select the participants?  This study recruited individuals across Poland via social media to complete an online survey. Study advertisements and the study website were shared via Facebook. (Materials and methods, Participants)

  1. Material and methods. Measures. "It includes five items (i.e.)...", why did you put i.e. if you mentioned the five items?
  2. In the following phrase: "To
    interpret the results, the properties characterising sten scores were also applied; sten". What did you refer to "sten"? In the Polish version of SWLS are stens.

A sten score indicates an individual's approximate position (as a range of values) with respect to the population of values and, therefore, to other people in that population. The individual sten scores are defined by reference to a standard normal distribution. Unlike stanine scores, which have a midpoint of five, sten scores have no midpoint (the midpoint is the value 5.5). Like stanines, individual sten scores are demarcated by half standard deviations. Thus, a sten score of 5 includes all standard scores from -.5 to zero and is centered at -0.25 and a sten score of 4 includes all standard scores from -1.0 to -0.5 and is centered at -0.75. A sten score of 1 includes all standard scores below -2.0. Sten scores of 6-10 "mirror" scores 5-1. The table below shows the standard scores that define stens and the percent of individuals drawn from a normal distribution that would receive sten score.

  1. Table 1: it's better to put a range (minimum, maximum) in the same row. For example: Age of young up to 35 years old (16, 35).

We have chanded with suggestion, see Table 1.

  1. Table 2. I don't understand the reason to put frequencies of all the points (1 to 7). A mean and standard desviation of each statement it's better. and more comprehensive as in the case by group I and II. What is it sten? I don't understand. 

We have changed the table. We have marked as table 2a. I suggest put all “huge tables 2,3,4 ” as the supplementary material.

  1. Table 3: The same as table 2. You have you provide the mean and standard desviation. The tables are so huge. We have changed the table. We have marked as table 3a.
  2. Table 4: why did you provide so much information? Can you divide into negative and positive attitudes in different tables, please? Or please, look for a simple way to understand the table.

 We have changed the table. We have marked as table 3a.

  1. Please English writing must improve. 

English hase been corrected . File is attached with English correction and below there is the last correction of new sentences Please see the corrected text.

Today, the many older people who are active after retirement are engaged in active ageing. According to WHO, “Active ageing is the process of optimizing opportunities for health, participation, and security in order to enhance the quality of life as people age” [4]. Active ageing applies to both individuals and groups within a population. It encourages people to realize their potential for physical, social and mental well-being throughout the life course and to participate in society.

Ageism is a type of discrimination that involves prejudice against people due to their age [5]. Similar to racism and sexism, ageism involves holding negative stereotypes about people of different ages. Gerontologist Robert N. Butler first used the term ageism to describe the discrimination of older adults. Today, the term is often applied to any age-based discrimination, whether it involves prejudice against children, teenagers, adults or senior citizens. Ageism is a highly prevalent, widespread phenomenon across many cultures.

Life satisfaction is defined as “a global subjective judgment concerning one’s life overall” [24]. Referring to a cognitive sense of satisfaction with life, life satisfaction involves a favourable attitude towards one’s life instead of an assessment of their current feelings only. In that dynamic, ample evidence indicates the impact of other factors of life satisfaction such as marital status, education, employment, income and being young [25].

Reviewer 3 Report

Thanks for recommending me as a reviewer. This study has been faithfully revised. 

Author Response

Thank you your comments.